# Expectations and attitudes towards medical artificial intelligence: A qualitative study in the field of stroke

Julia Amann[1]*, Effy Vayena[1], Kelly E. Ormond[1], Dietmar Frey[2], Vince I. Madai[2,3,4], Alessandro Blasimme[1]*

1 Health Ethics and Policy Lab, Department of Health Sciences and Technology, ETH Zurich, Zurich, Switzerland, 2 CLAIM—Charité Lab for AI in Medicine, Charité Universitätsmedizin Berlin, Berlin, Germany, 3 QUEST Center for Responsible Research, Berlin Institute of Health (BIH), Charité Universitätsmedizin Berlin, Berlin, Germany, 4 School of Computing and Digital Technology, Faculty of Computing, Engineering and the Built Environment, Birmingham City University, Birmingham, United Kingdom

* julia.amann@hest.ethz.ch (JA); alessandro.blasimme@hest.ethz.ch (AB)

## Abstract

### Introduction

Artificial intelligence (AI) has the potential to transform clinical decision-making as we know it. Powered by sophisticated machine learning algorithms, clinical decision support systems (CDSS) can generate unprecedented amounts of predictive information about individuals' health. Yet, despite the potential of these systems to promote proactive decision-making and improve health outcomes, their utility and impact remain poorly understood due to their still rare application in clinical practice. Taking the example of AI-powered CDSS in stroke medicine as a case in point, this paper provides a nuanced account of stroke survivors', family members', and healthcare professionals' expectations and attitudes towards medical AI.

### Methods

We followed a qualitative research design informed by the sociology of expectations, which recognizes the generative role of individuals' expectations in shaping scientific and technological change. Semi-structured interviews were conducted with stroke survivors, family members, and healthcare professionals specialized in stroke based in Germany and Switzerland. Data was analyzed using a combination of inductive and deductive thematic analysis.

### Results

Based on the participants' deliberations, we identified four presumed roles that medical AI could play in stroke medicine, including an administrative, assistive, advisory, and autonomous role AI. While most participants held positive attitudes towards medical AI and its potential to increase accuracy, speed, and efficiency in medical decision making, they also cautioned that it is not a stand-alone solution and may even lead to new problems. Participants particularly emphasized the importance of relational aspects and raised questions

**Data Availability Statement:** All relevant data are within the paper and its Supporting Information files, i.e., translated excerpts of the transcripts relevant to the study are available within the paper.

**Funding:** This research has received funding from the European Union's Horizon 2020 research and innovation programme under grant agreement No. 777107 (PRECISE4Q), PI: DF. Funder website: https://ec.europa.eu/ The funders had no role in study design, data collection and analysis, decision to publish, or preparation of the manuscript.

**Competing interests:** The authors declare no competing interests.

regarding the impact of AI on roles and responsibilities and patients' rights to information and decision-making. These findings shed light on the potential impact of medical AI on professional identities, role perceptions, and the doctor-patient relationship.

## Conclusion

Our findings highlight the need for a more differentiated approach to identifying and tackling pertinent ethical and legal issues in the context of medical AI. We advocate for stakeholder and public involvement in the development of AI and AI governance to ensure that medical AI offers solutions to the most pressing challenges patients and clinicians face in clinical care.

## Introduction

There is an ever-increasing hype surrounding the use of artificial intelligence (AI) to catalyze personalized medicine to the next level. Yet to date, there is little concrete evidence on how medical AI will impact patient care and health outcomes in the long term. A core area of AI application in medicine is in clinical decision support. Clinical decision-support systems (CDSS) are interactive software systems designed to help clinicians focus their attention [1] and support them in decision-making tasks, such as diagnosis or treatment recommendations [2]. They can be used to improve outcome prediction, prioritize treatment or help in identifying the best course of action for a specific patient [1]. What distinguishes AI-based CDSS from other types of CDSS is that they are not knowledge based, i.e. they do not rely on a manually pre-defined given set of rules based on expert medical knowledge but instead leverage AI to *learn* from data input to generate predictions or recommendations [3].

Commitment to medical AI has largely been spurred by its promises rather than its actual contributions to clinical practice. Indeed, evidence suggests that despite their attested performance and technical sophistication [4,5], AI-powered CDSS, particularly those based on the latest deep neural networks, often fail to realize their potential beyond the proof-of-concept stage [6–8]. This translational gap makes it difficult to determine clinical utility and impact of medical AI in practice [1,9]. A main reason for this can be seen in the slow uptake and adoption of AI-powered tools due to regulatory uncertainty, organizational challenges, and attitudinal barriers [1,8,10–12]. Moreover, while the scholarly debate on the potential risks and benefits of medical AI is in full swing [13–17], the views of prospective users and beneficiaries of these novel technologies are often missing from the picture and consequently disregarded when it comes to product design and governance structures [11,18–21].

Taking stroke, a highly prevalent and devastating disease, where AI-based CDSS could lead to significant clinical benefits, as a case in point, we aim to address this gap in the current literature. The primary aims of this study were to explore the views of stroke survivors, their family members, and healthcare professionals specialized in stroke regarding the use of stroke related medical AI. More specifically, we aimed to elicit their expectations and attitudes towards stroke related medical AI, focusing on the perceived benefits and risks when applied in the clinical setting. By recognizing that expectations shape how people interact with technology, and they influence the ways in which it is used, this study goes beyond the question of mere technology acceptance or rejection. Instead, it tells us something about how medical AI will likely be applied in practice and which considerations or guidelines are warranted to ensure the ethical, safe, and efficient use of these technologies. This paper complements existing

normative guidelines and ethical frameworks [22–26] by empirically probing stakeholders' views on ethically relevant issues relating to the use of medical AI in a concrete clinical context, namely, stroke medicine.

## Methods

### Study design

We adopted a qualitative research design, using the sociology of expectations [27,28] as our theoretical framework. The sociology of expectations is a widely used framework in the field of Science and Technology Studies (STS) that recognizes the generative and performative role of individuals' expectations in shaping scientific and technological change [27–29]. It postulates that the successful development and diffusion of novel technologies heavily depends on people's expectations about their future capabilities and potential. For the purpose of this study, we conceptualized *views* to encompass attitudes, opinions, beliefs, feelings, understandings, experiences, and expectations [30]. This paper follows the Consolidated Criteria for Reporting Qualitative Research (COREQ) reporting guideline for qualitative research studies [31]. The study was approved by the ETH Ethics Commission (EK 2019-N-88). Written informed consent was obtained from all participants.

### Recruitment

Semi-structured interviews were conducted with stroke survivors, family members of stroke survivors, and healthcare professionals specialized in stroke based in Switzerland and Germany. To be eligible to participate in this study, individuals needed to belong to the study population (stroke survivor, family member of stroke survivor, healthcare professional specialized in stroke), be above 18 years of age, and provide written informed consent. We adopted a convenience sampling approach combined with snowball sampling. Specifically, we relied on our professional network and contacted medical professional associations and patient associations to support our recruitment efforts by advertising the study through their websites and mailing lists.

### Data collection

Semi-structured interviews were chosen as a means of data collection because they allow for an in-depth exploration of individuals' personal stories, reflections, and reasoning [32]. The interview guide (Appendix 1) consisted of three sections and was pre-tested with two healthcare professionals and a family member of a stroke survivor. The first two sections aimed to explore participants' experiences with stroke and stroke care and to assess how they conceptualize AI and its application in healthcare. We then provided participants with a short vignette (Appendix 2), describing how such an AI-based CDSS would be used in practice, using a fictional patient as an example. Immediately after, the third section explored participants' expectations and attitudes towards AI-powered CDSS for stroke. We were particularly interested in participants' views regarding relevant ethical issues and how they might be affected by medical AI, acknowledging that their responses would primarily stem from personal views and moral intuitions rather being theoretically founded. In seeking to encourage participants to express themselves freely, we also did not correct them when we uncovered misunderstandings or misconceptions of what medical AI means (e.g., when participants talked about possible areas of applications or benefits that would, in fact, not depend on AI-based solutions). If not brought up by participants themselves, we prompted them with follow-up questions to reflect on these issues. While AI is used as an umbrella term throughout the manuscript, it is important to note that the way both the interview questions and the vignette were phrased, AI in this

context referred to deep learning/neural network with their black box character. The term artificial intelligence was used to keep the interview language simple.

Interviews were conducted between September 2019 and February 2020 either face-to-face or on the phone, according to the participants' preferences. All but two interviews were conducted by the first author (JA), a health communication researcher, who is a native German speaker trained in qualitative research methods. Two interviews were conducted in Italian by a native Italian-speaking research assistant in presence of the first author, who has a working understanding of Italian. All Interviews were audio-recorded and transcribed verbatim for analysis.

### Data analysis

In total, we conducted 34 interviews with 35 participants. Two couples participated in the study, one of which was interviewed together, and one triad of participants, interviewed separately. One stroke survivor interview had to be excluded as the participant diverted strongly from the questions. The remaining 33 interviews lasted between 22 and 78 minutes, 40 min on average. A total of 22 hours and 38 minutes of audio material was transcribed for analysis. One interview was conducted with a patient with mild to moderate aphasia, which rendered transcription of the interview challenging; for this transcript, we thus drew on field notes taken by the interviewer (JA) during the interview.

All interviews were analyzed in the original language (German/Italian). The analysis was led by the first author (JA) and involved a research team with expertise in bioethics (AB, EV, KO) and genetic counseling (KO). Data was analyzed using a combination of inductive and deductive thematic analysis in MAXQDA [32]. We started by familiarizing ourselves with the data through an in-depth review of all transcripts accompanied by the respective recordings. One author (JA) conducted a first round of line-by-line coding and extensive memo-taking both in written and visual form to engage with the data on a more abstract level [33]. In a next step, two researchers (JA, AB) merged codes into overarching themes and subthemes in an iterative process. Guided by our research objective, we examined the codes and associated data excerpts to uncover patterns across the data which we merged into candidate themes. The entire research team then reviewed our candidate themes and all the relevant coded data for each of them to ensure that themes were meaningful, internally coherent, distinctive and, yet, related to one another. In the final analytic phase, we named the themes, drafted theme descriptions, and identified interview excerpts to present. Supporting quotes presented in this paper have been translated to English and are accompanied by anonymized participant identifiers.

## Results

### Participant characteristics

Of the 34 individuals who took part in this study, 14 (41%) were healthcare professionals (HCP), 14 (41%) were stroke survivors (Pat), and 6 (18%) were family members of stroke survivors (FM), two of whom were also healthcare professionals by training (FM3, FM4). While none of the participants mentioned having had direct experience with medical AI in their work/treatment, some mentioned having heard or read about possible applications. Table 1 presents the participants' characteristics.

### Expectations and attitudes towards AI in stroke medicine

We identified three overarching themes and subthemes: 1) presumed roles of AI in the clinical setting, 2) perceived opportunities and limits of medical AI, 3) perceived challenges, risks, and open questions (Fig 1).

**Table 1. Participants' characteristics.**

| Healthcare professionals (HCP) (N = 14) | | Stroke survivor (Pat) (N = 14) | | Family members (FM) (N = 6) | |
|---|---|---|---|---|---|
| **Professional Role, n (%)** | | **Age (mean, range)** | 62 (47–80) | **Role, n (%)** | |
| Physician | 7 (50) | | | Partner | 5 (83) |
| Occupational therapist | 1 (7) | | | Child | 1 (17) |
| Physiotherapist | 4 (29) | | | | |
| Neuropsychologist | 2 (14) | | | | |
| **Gender, n (%)** | | **Gender**, n (%) | | **Gender, n (%)** | |
| male | 5 (36) | male | 9 (64) | male | 1 (17) |
| female | 9 (64) | female | 5 (36) | female | 5 (83) |
| **Years in practice, n (%)** | | **Time since stroke, n (%)** | | **Time since stroke, n (%)** | |
| <1 year | 4 (29) | <1year | 4 (29) | <1year | 0 (0) |
| 1–5 years | 5 (36) | 1–5 years | 2 (14) | 1–5 years | 1 (17) |
| 6–10 years | 0 (0) | >5years | 8 (57) | >5years | 5 (83) |
| >10 years | 5 (36) | | | | |

**Presumed roles of AI in the clinical setting.** Our findings suggests that there was no common understanding of what medical AI is and what it can or cannot do among our study participants. Based on the interviewees' deliberations, we identified four presumed roles along a continuum that medical AI could play in the stroke medicine: a) an administrative assistant; b) a clinical decision support system; c) the clinician's right hand (advisory); d) fully autonomous medical AI (see Table 2). It should be noted that the boundaries between the presumed roles we identified are not perfectly distinct but understood on a continuum. HCPs tended to describe a wider range of roles, with 9 out of 14 describing at least 3 roles. Stroke survivors and family members, on the other hand, were more likely to describe one or two roles.

*AI as an administrative assistant.* On the one hand, interviewees considered that AI-powered systems could assume the role of an *administrative assistant*, taking over repetitive and mundane tasks to free up clinicians' time, leaving them more time for other, more demanding tasks. These tasks would mainly be administrative in nature and may, for instance, include patient documentation, data synthesis, or monitoring patient vitals. Participants suggested this type of AI-powered systems may be able to enhance interprofessional communication and the flow of information across different settings, including the clinical encounter and doctor-

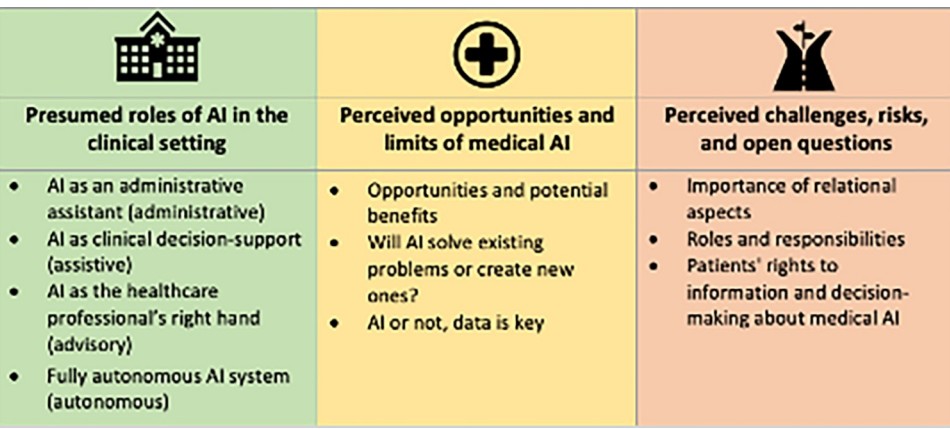

**Fig 1. Overview of themes and subthemes.**

**Table 2. Presumed roles of AI in the clinical setting.**

| | Data collection & synthesis (administrative) | Clinical decision support (assistive) | Clinician's right hand (advisory) | Fully autonomous AI system (autonomous) |
|---|---|---|---|---|
| | AI system provides comprehensive patient data and facilitates communication and data flow | AI system provides protocol-like structure, as well as options to be interpreted/weighed by the clinician | AI system advises clinicians and can complement their expertise and provides recommendation that need to be actively rejected by the clinician | AI system identifies course of action autonomously and the clinician can no longer (easily) intervene, incl. application of AI in robotics |
| | *"I can imagine that in a much more general way, artificial intelligence could increase patient health, if done more comprehensively. For example, by collecting risk factors and everything and providing, for example, a quick tool [with this information] for the family doctor, I don't know, or for the specialist who is faced with a patient that maybe they know little about." (HCP10)* | *"The lowest level would be alerts, like decision trees in the emergency setting. So, [the program prompts me] the patient has this and this impairment, have you thought about this and this, then I respond to it, and then comes the next suggestion, the program recommends this examination, and of course I can always intervene." (HCP1)* | *"It's just like getting a second doctor's opinion. A second [doctor] perhaps examines the clinical picture differently. There [in an AI system] is a lot of information from many stroke patients, and that makes it easier for a doctor to make a decision, I think, especially perhaps for doctors who otherwise don't engage in continuous training." (Pat8)* | *"I could even imagine that in the distant future robots will perform surgeries [...] There are already surgical robots that translate th surgeon's movements and simply guarantee more precision. [...] I can imagine that the surgeon will be replaced by artificial intelligence, at least to perform less complicated surgeries, standard procedures." (Pat2)* |
| HCP (N = 14) | 10 (71.4%) | 10 (71.4%) | 11 (78.6%) | 9 (64.3%) |
| Stroke survivors (N = 14) | 6 (42.9%) | 5 (35.7%) | 7 (50.0%) | 8 (57.1%) |
| Family members (N = 6) | 3 (50.0%) | 3 (50.0%) | 0 (0%) | 4 (66.7%) |

patient communication. By making information available at the point of care and freeing up time, it would allow providers to work at the top of their scope.

*AI as a clinical decision support system.* Other accounts of AI-powered systems underlined their role as a *clinical decision support system*, providing clinicians with a structure or protocol. Participants suggested that this could help clinicians to adhere to guidelines in a check-list manner. Perceived benefits would be that the system could point their attention to aspects they may have overlooked, and to facilitate faster and more accurate diagnosis by analyzing greater amounts of data in less time. One participant suggested that this could, in fact, also reassure clinicians because they would be basing their decisions on more robust data. Moreover, participants viewed an AI-based system as a possible source of inspiration on what to try out, specifically in the rehabilitation context, to provide tailored treatment to patients and to identify attainable rehabilitation goals (akin to the idea of a digital twin [34]).

*AI as the clinician's right hand.* While many participants stressed that clinicians could and should interpret the system's output (and should not blindly trust the system's recommendations), many considered that AI could expand or complement clinicians' expertise and capabilities in an advisory function. Such accounts suggest the role of *AI as the clinician's right hand*, or in other words, a complement to or extension of the healthcare professional. This type of tool could guide clinicians when presented with particularly complex cases (e.g., orphan diseases) or in time sensitive or resource constrained settings, such as the ICU, overnight shifts, or in case of staff shortage. In this way, it could assist less experienced or well-trained clinicians in the decision-making process, or even act as a second opinion. These findings suggest it is unlikely that AI would replace expert clinicians, but rather help to compensate for lack of time or expertise in resource-constrained settings. Moreover, participants cautioned that if considered an advisor to the clinician, system recommendations might less likely be challenged by clinicians.

*Fully autonomous medical AI-systems*. While frequently mentioned as a potential use of AI, few participants viewed *fully autonomous medical AI-systems* as realistic in the near future. In this context, robotics (e.g., care robots, surgery robots or therapy robots) was the most frequently mentioned area of application in the clinical setting. During the interviews, some participants occasionally referred to "decisions made by the AI". However, it is unclear whether these implicit attributions of decision-making power were intentional or not. One clinician, for instance, described how an AI system could not only help to exclude differential diagnosis but could also take on tasks like ordering examinations. From this description it was unclear whether, in the view of the participant, any intervention from the clinician was foreseen to approve the AI's recommendations.

### Perceived opportunities and limits of medical AI

The second theme we identified captures participants' views on the opportunities and potential benefits of AI-powered CDSS whilst at the same time reflecting skepticism towards AI solutionism.

**Opportunities and potential benefits of medical AI.** Most participants had a positive attitude towards medical AI and echoed the commonly promoted *opportunities and potential benefits* of medical AI that are reiterated in the public and academic discourse: accuracy, speed, and efficiency. Some participants explicitly linked these perceived benefits to an increase in patient safety and quality of care.

*"I think this [medical AI] is good, because it means that data can be collected again for the future, which can make any programs more precise, which can set the focus more precisely for an evaluation and then a recommendation as to what is good for the individual in order to get well again [after suffering a stroke]." (Pat8)*

In addition, some participants further suggested or hoped that AI may lead to more accessible, objective and consequently fair healthcare as it may compensate for human bias, healthcare providers' preconceptions about certain individuals, or lack of well-trained staff.

*"I just hope from something like that, so from computer programs or algorithms actually, that assumptions that we, I think, always make in everyday life as humans—because someone is old or somehow looks like that or is old on paper—will be less incorporated [in the decision-making process], so this subjectivity." (HCP13)*

When talking about the potential gains in objectivity AI-based CDSS might offer, a clinician reflected further on fairness and how it may impact the decision-making process:

*" I don't want to accuse anyone of consciously using more diagnostics or less diagnostics in a certain way because of sympathies or something like that, or because of some conscious decision about how vital someone seems, but there is perhaps a danger, and this is where artificial intelligence could help to make things fairer, so to speak. The question is whether it is always more humane. [. . .] if we consider justice, we have quite different definitions, which makes it more difficult again, so when does someone have a right to certain things and when rather less. So, is it fair, if someone has a genetic risk, he should get more than if someone gets the stroke maybe because of supposedly self-inflicted things? But there again the question is, with a nicotine addiction, for example, to what extent is there a free will at all? And to what extent is not also the addiction determined by some genetic aspects or environmental factors, where the patient can't do anything about it?" (HCP5)*

**Will AI solve existing problems or create new ones?.** Several participants cautioned that AI should not be hailed as a panacea that will solve all of healthcare's problems, raising the question: *Will AI solve existing problems or create new ones*? Some suggested that AI might, in fact, not change all the much about current practices as it was just a new tool that would not impact principles such as patient autonomy, justice, or trust. Others expressed concern that medical AI might not be able to address some of the existing "most basic" challenges faced in stroke medicine and healthcare more generally, e.g., economic pressure, existing inequalities, and changing individual health behaviors:

> *"We're not bad at high-tech medicine, but we can't get a grip on normal medicine. [. . .] The normal being the care of the population with family doctors, with specialists–it's a disaster." (Pat1)*

> *"I would rather assume that the problem is not that you have the wrong options [provided by the AI system], but rather that you generally lack the resources to properly implement the options that are available. So, for example, sufficient physiotherapy in the outpatient area or something like that. That a computer-aided decision or simulation of different options would not change anything about the problem that already exists. would not change the problem that already exists." (HCP4)*

Interviews with stroke survivors and family members were dominated by personal narratives and challenges faced in the aftermath of stroke, ranging from bureaucratic hurdles (e.g. negotiations with insurers) to interpersonal challenges (e.g. activities of daily living, such as getting dressed or pursuing a professional career) and mental health problems that they did not expect medical AI would change.

A priority all three stakeholder groups voiced was the need for access to relevant patient information at the point of care to improve efficiency, treatment, and patient safety. Having to provide information repeatedly was described as a particularly frustrating experience by patients and family members.

> *"I think it's completely unnecessary to have to fill out a questionnaire at every new doctor's office, I think it's just redundant, because the data is stored at every doctor's office anyway, so you can actually store it right on the card and have it read out there and then." (FM1)*

Finally, some participants expressed concerns that AI may introduce new problems.

> *"Well, one shouldn't overestimate AI, I have a feeling. It's not the solution.. [. . .] Nobody is thinking, should we really do this? Do we need to do that? And what are the long-term consequences? And that's where I think we tend to go too far, especially in healthcare, and by [introducing] potential solutions or improvements often we create new problems, which you can't really anticipate." (HCP0)*

**AI or not, data is key.** Most participants considered comprehensive patient data pivotal to improving healthcare services in stroke medicine, whether an AI system was involved in the collection or analysis of such data seemed to be secondary. While some considered that AI-powered systems could help to collect, synthesize, and manage patient and research data, others did not explicitly mention or did not believe that AI would be needed for these types of tasks.

*"Digitization leads to a more barrier-free flow of data, but that's not necessarily artificial intelligence, but it is about the precondition that information is used in such a way that it benefits the individual patient or the doctor or the nurse because it provides more information or processes it in such a way [. . .]." (HCP1)*

Only one of the participants cautioned against what he negatively referred to as *data faith*, arguing that generating more data might also lead to insecurities and may in fact not be the solution to improving healthcare.

*"The more data we have, the greater the uncertainty. And I think the big problem we have now is simply this, too much faith in data, or thinking that if we just collect enough data, we can improve everything. I think we should just shift down a gear there." (Pat2)*

## Perceived challenges, risks & open questions

We identified three key areas of perceived challenges, risks, and open questions related to the adoption of AI in clinical practice, namely a) importance of relational aspects, b) roles and responsibilities, c) patients' rights to information and decision making.

**Importance of relational aspects in healthcare.**   *Relational aspects in healthcare* featured heavily in our interviews, especially in the context of rehabilitation. Many participants considered that the importance of trust in clinical care would ultimately shape the impact of AI on the doctor-patient relationship and medical decision-making. More specifically, most were convinced that AI would not be able to replace humans in healthcare, because healthcare relies on human relationships, empathy, and human warmth. Stroke survivors in particularly highlighted the healing, therapeutic effect of having someone to talk to and feeling understood.

*"There was a phase when I wasn't doing well psychologically, and I was just happy to be able to talk to someone. And that is also good for the language, to talk about it in depth. And that was very important. To withdraw and if you only do that with the AI, I don't see that." (Pat12)*

Stroke survivors and clinicians voiced concerns that the adoption of AI in clinical practice might slowly lead to dehumanizing healthcare, steering healthcare professionals to lose sight of the person in front of them and having patients worry about becoming just a number in the system.

*"I can imagine that there is a danger that health professionals will rely more on artificial intelligence and perhaps fixate on it and pay less attention to the patients and their wishes." (HCP8)*

Some participants were also convinced that efficiency gains owed to AI would not directly benefit patients but instead lead to further streamlining i.e., a reduction in healthcare personnel. Sharing his frustration, one of the stroke survivors criticized trends of outsourcing interpersonal tasks that are central to healthcare to volunteers, painting a grim picture of the future of healthcare:

*"[When volunteering], I see again and again how much this human contact is needed and desired, and I also see the efforts in healthcare to automate more and more to save personnel*

*costs, and to shift this task to volunteers. [. . .] Here everything is being streamlined and I can't imagine more time being devoted to the patient because some of the work is being shifted to artificial intelligence. [. . .] If we think about the developments over twenty or two hundred years into the future, it's actually going to be Starship Enterprise, that the patient lies in a sterile room and if he's lucky he gets to see a human being once a day or once a week." (Pat2)*

**Roles and responsibilities.** Losely related was the subtheme *roles, and responsibilities* which captured participants views on some of the central topics regarding the practical implementation of AI, including questions of decision autonomy, transparency, responsibility, and fairness. Underscoring the clinician's epistemic authority, most of the participants agreed that decision-making power should rest with clinicians and patients, and that humans should always be able to intervene, whether to confirm or to correct the AI system's output.

*"I think that as a therapist you should still have the possibility to intervene to be able to correct certain errors. And that you can take this, which has now been taken out of thousands, that you can still individually adapt it to the patient, that you can take it as evidence, but then also incorporate personal experience and intuition, to create and implement the special rehabilitation program for the patient." (HCP9)*

Going a step further, some participants cautioned that AI-powered systems may lead clinicians to second-guess themselves, doubt their own intuition, or even blindly trust the system thus neglecting their medical responsibility. Some also considered that overreliance may lead to a loss of expertise and competences in future generations of clinicians and, consequently, dependence.

*"Yes, definitely. So, I think you also have to look, of course, that the treatment team then also does not rely one hundred percent on artificial intelligence, but always questions that again or also listens to the family." (FM1)*

*"[A risk is] that it affects the training of physicians in such a way that in the end everybody relies on it, and nobody dares to stand up and have their own opinion. That is now also generally the problem, which sometimes strikes me. When we are actually completely sure what the clinical diagnosis is, we still do an MRI in so many cases where it is not necessary, because we somehow don't trust ourselves [. . .]. I can imagine that this would also be a risk that we sell ourselves short, or at some point also have to, because we really can no longer do it." (HCP6)*

Related to considerations of decision-making power, medical error and responsibility were discussed. While some participants were hopeful that AI systems could reduce or prevent medical error, others recognized that both humans and AI can err, highlighted potential sources of error, like biased data sets or errors during data entry.

*"And either way, mistakes can happen, so the computer system can also somehow give a recommendation that is ultimately wrong, because some parameters were entered incorrectly, because it makes a mistake." (HCP4)*

There was wide agreement that the responsibility for medical decisions alongside the decision-making authority presently rests with the healthcare professional or care team. While some participants advocated that at least a rudimentary understanding of the AI system would

be necessary to ensure quality and patient safety, others described the lack of explainability as not unique to medical AI, providing examples of "black box" tools currently in use.

*"One must not relinquish one's responsibility. At the same time, this is already happening, for example, with this CT monitoring. I would never check what this program has presented to me. But I don't know of any case where there was a problem or the patient had an arrhythmia after all, and then it turned out afterwards that the program didn't identify it. But in this respect, I actually defer my responsibility to the reading of the monitoring. Not responsibility, because if there is an error, then I still have to take the blame. But I have no way of verifying that [the output]. Responsibility is definitely on the doctor." (HCP4)*

Yet, some participants also acknowledged that forms of shared or developer responsibility might be possible in case of a malfunctioning system, for example.

*"In my eyes [the doctor bears] the responsibility, yes, who else should it be? Well, we can't hold an IT specialist responsible. We can only hold him responsible if he programs a system error or something." (Pat7)*

**Patients' rights to information and decision-making about medical AI.** The third sub-theme we identified captured participants' views concerning *patients' rights to information and decision-making*, reflecting considerations related to patient autonomy, data protection, and privacy. Many participants explicitly stated that patients (or their designated proxies) should have a say when it comes to data collection and medical decision-making, one proposing organ donation as an analogy.

*"I think, there must be very clear information, what the programs can do and where the limits are and now comes again so the question, I do not know now, what is already discussed there, I think that must be quite transparent. And then it must be a clear consent, so as I say as a patient or as a person, okay, I agree to my organs being donated or so, it must also apply to such a situation that I agree to be tracked and I must also be able to revoke that." (FM6)*

In contrast, only few pointed to potential limits of patient autonomy when it comes to the choice of tools and resources used by clinicians.

*"You don't ask the patient whether the doctor is allowed to use computers or what literature he reads or is allowed to read. I can't imagine that, so I think the doctor must be free to decide which procedures he uses, and the patient shouldn't be allowed to have a say in that either." (Pat2)*

There was some disagreement on whether patients should be informed if an AI system was used to guide their care. Some argued that informing patients was a (moral) obligation, like informing about medication side effects, and could promote trust.

*"Probably it does make sense to inform the patients that the prediction that has been made, how that has been generated so roughly. So, what patient data from how many patients went in there and normally that would be evaluated yes probably how good the evaluation is, what it has shown, how reliably the program can predict what, what errors are possible there. It probably makes sense [to inform the patient] like how you have to inform about side effects of*

*a drug, to give a brief information overview, if you give a clear prognosis, so to speak, on what it is based on." (HCP3)*

Others considered disclosure unnecessary or simply challenging, concluding that too much information might in fact be harmful, confuse and overwhelm patients, or even erode their trust, especially in the acute setting. Reiterating the importance of trust in the doctor-patient relationship, one participant emphasized that information should not be overrated.

*"Well, I am of the opinion that today this demand for absolute information is a bit stupid. So, I must say, in such a case or otherwise also in normal life, the word trust is something that is very, very important." (Pat7)*

Several participants voiced concerns regarding data sharing, also independently of AI, including the risk of surveillance, loss of privacy, and information getting into the wrong hands, like hackers, banks, employers, or insurance companies. However, there were also a few participants who either did not attribute much value or concern to data protection and privacy or simply considered it an illusion. Participants often tended to weigh the risks of data sharing and subsequent use of data for medical AI against potential personal or public health benefits.

*"FM5: There are people in our circle of acquaintances for whom data protection is very important. We're not quite one of them, for us it's simply more important that the data is processed in a way so that it can be used for something. We are both no longer in professional life. That means that this could not be exploited by the employer, who then says, I now know about you, you have this and that, we'd better separate, so in this respect, I don't know why I would have to be afraid. What do you say, [name of partner]?*

*Pat13: Yes, I feel the same way. Above all, we want to serve the success of the cause."*

## Discussion

Prior research on medical AI has investigated factors related to technology acceptance and AI adoption, mostly from the clinicians' point of view [35–39]. This is, to our knowledge, amongst the first studies to investigate the ethical, legal, and societal implications of medical AI from the perspectives of patients, family members, and HCPs. Our findings draw attention to some currently under-investigated issues, namely: 1) the need for a more nuanced understanding of what medical AI entails, 2) the implications of medical AI for professional identity, role perceptions, and the doctor-patient relationship, and 3) the need for challenging AI solutionism and promoting stakeholder engagement. Even though this study was carried out in the field of stroke medicine, there are some general patterns that are likely transferable to other clinical settings.

### Moving towards a more nuanced understanding of medical AI

In the academic and policy discourse, medical AI is often used as an umbrella term, lumping together various types of applications irrespective of their envisioned role in the decision-making process. In this sense, a range of AI-based CDSS are often treated as equal, rather than distinguishing between different types and purposes of AI-based applications [40]. In contrast, we found that AI-based CDSS were understood on a continuum, ranging from the management of administrative tasks to fully automated clinical decisions. It is reasonable to assume that different implementations of AI in healthcare correspond to a more nuanced articulation

of its social, ethical, and legal implications on the part of stakeholders [27,41]. This, in turn, highlights the need for a more nuanced approach to identifying and addressing pertinent ethical and legal issues, like liability, justice, and explainability, as they may vary across the different types of medical AI. Taking the prominent topic of explainability as an example, one can observe that the academic discourse is largely characterized by a debate on whether explainable AI is important or at all possible, especially in the case of deep neural networks but also sophisticated applications of more classic approaches such as ensemble methods for decision trees [42–45]. However, such debates revolve around a relatively limited array of uses–mostly AI-driven diagnosis based on clinical images. Lesser attention is paid to how the issue of explainability plays out in different contexts, including allegedly more trivial ones like administrative tasks. With real-world examples of AI bias leading to discrimination of vulnerable and disadvantaged populations [24,46,47], it is evident that precautionary measures need to be put into place to ensure that medical AI benefits all of society rather than increasing existing inequities.

## Implications for professional identity, role perceptions, and the doctor-patient relationship

The type of role that clinician and patient assign to the medical AI-system in the decision-making process will likely determine not only what they expect from the system but also how they interact with it and one another [41]. This raises the fundamental question of how medical AI will impact professional identities, role perception and, ultimately doctor-patient relationships —issues we believe are currently under investigated [48,49].

Our findings are consistent with earlier work [50–52], suggesting that while many stakeholders recognize the potential of medical AI, there are concerns that it will lead to overreliance, deskilling, and a dehumanization of care. These considerations have direct implications for clinicians' role perceptions. Authors like Verghese and colleagues advocate for a partnership between physicians and AI, postulating that "a well-informed, empathetic clinician armed with good predictive tools and unburdened from clerical drudgery—can bring physicians closer to fulfilling Peabody's maxim that the secret of care is in 'caring for the patient'". [53] But what would this mean for the physician's professional identity and what might patients expect from their physicians?.

Our findings highlight the fear of a technology-induced dehumanization of care, emphasizing the importance of relational aspects and empathy in stroke medicine and healthcare more generally, mirroring earlier work [54–56]. The role of interpersonal relationships is also discussed in an emerging body of literature, which suggests that the AI revolution will, in fact, require a shift towards a more relationship-oriented role of clinicians [49,53,57–59]. Nagy and colleagues, for instance, argue that even though AI may unburden clinicians from some of the analytical and administrative demands–as also suggested by findings of our study—it might, on the other hand, increase the interpersonal demands of patient care [49]. In line with others, the authors suggest that medical AI may identify a greater number of viable treatment options than a physician would, leading to an increase in information regarding prognosis and the potential risks and benefits associated with all those different treatment options. Resultantly, clinicians would have to dedicate more time to informing the patient and eliciting their values and preferences, handling areas of uncertainty, balancing competing risks, and finally, supporting patients emotionally.

In addition to highlighting the need and desire for a more relationship-oriented role, our findings also indicate that clinicians are expected to maintain their responsibilities as epistemic authority. In other words, according to our participants, clinicians should remain in charge of interpreting and acting on recommendations generated by AI-powered CDSS and should also

be the ones held accountable. Contrary to our participants' views on decision-making power and responsibility, some authors have argued that in case of its superior performance in clinical tasks, clinicians would have an epistemic obligation to defer to the AI system's recommendations or align their judgements [41,60]. However, given that clinicians remain the ones expected to be morally and legally responsible [61,62], it seems unlikely that they will delegate decision-making power just yet.

A look at the literature suggests that much of the current discourse around AI ethics focuses on questions of data protection and privacy, liability, and fairness (bias). While these are undeniably crucial issues, we argue that they can, at least to a certain extent, be addressed by regulatory measures. The impact of medical AI on professional identities and role perceptions, on the other hand, goes to the core of what it means to be a clinician caring for patients. To date, only few authors have addressed questions of professional identity beyond the fear of job-loss and technology acceptance [35,63]. We believe that these humanistic aspects of professionalism deserve further attention in research and medical education. When preparing clinicians for the ethical challenges of medical AI [64], it will be paramount to focus on strengthening empathy, interpersonal, and communication skills.

## The need for challenging AI solutionism and promoting stakeholder engagement

Our study captured the tension between the perceived benefits and participants' skepticism towards AI-powered CDSS. Indeed, we found that even though participants were cognizant of potential opportunities, like increased efficiency, many opposed the idea of AI solutionism. Participants identified three areas of concern, which addressed relational aspects, professional roles and responsibilities, and patients' rights to information and decision making. We also observed that participants' narratives were very much anchored in the real-world challenges they experienced. Many of the possible applications participants referred to would effectively not require AI as also pointed out by some of the participants themselves. Instead, these applications would simply rely on data accessibility and interoperability. Based on what we know about the fragmented healthcare data system, it was therefore not surprising that a core challenge participants wanted to see addressed was data availability and accessibility. The recurrent narrative of challenging AI solutionism we found is in stark contrast to the media and political hype around medical AI which also dominates the scientific landscape, apart from a few noteworthy exceptions [65–67]. AI solutionism refers to the assumption that "as long as there is enough data, any human outcome can be computed based on machine learning algorithms" [68]. Our findings suggest that instead of relying on cutting-edge technologies to miraculously solve all of healthcare's problems, it might be more appropriate to first and foremost understand the needs and perceived priorities of those most directly affected.

A first step towards this goal is involving stakeholders including the wider public both in the development and in the governance of medical AI [22,69–71]. From an intuitive and normative stance, involving those most directly affected by medical AI seems reasonable and adequate. But there is also empirical evidence highlighting the advantages of stakeholder involvement. A recent review, for instance, has shown that stakeholder engagement in developing AI ethics guidelines led to more comprehensive ethical guidance with greater applicability [21]. However, the study also showed that presently only 38% of AI ethics guidelines reported some form of stakeholder engagement with even fewer of them providing information on the engagement process. We support the authors' call for meaningful and transparent stakeholder and citizen engagement to devise more sustainable, inclusive, and practical ethical guidance. While acknowledging the challenges that stakeholder engagement and the

involvement of vulnerable and socially disadvantaged groups, in particular, can entail, we urge AI-developers and policymakers to enable an accessible and inclusive dialogue [72].

## Strengths and limitations

Findings of this study should be interpreted considering some limitations. We do have to consider the possibility of a selection bias, i.e., that our sample included more technology-savvy participants who may have had more favorable attitudes towards medical AI in stroke medicine. It is also important to note that we completed data collection just before the Covid-19 pandemic. Given that the pandemic has put public health and medical ethics onto the public agenda, it is possible that different stakeholder groups may have become more aware of ethical issues surrounding medical AI. Finally, we prompted participants to reflect on certain ethical issues if they did not address them on their own. While this may have steered the conversation in certain ways, we believe that it helped to stimulate participants' reflections, particularly for those less familiar with the interview setting or research topic.

## Conclusions

This is, to our knowledge, one of the first studies to investigate the ethical, legal, and societal implications of medical AI from the perspectives of patients, family members, and healthcare professionals. Our findings highlight the need for a more differentiated approach to identifying and tackling pertinent ethical and legal issues in the context of medical AI. Our work also sheds light on the implications of medical AI for professional identity and role perceptions and the need for challenging AI solutionism in healthcare. We advocate for stakeholder and public involvement in the development of AI and AI governance to ensure that medical AI offers solutions to the most pressing challenges patients and clinicians face in clinical care. Here, we urge AI developers and policymakers to pay particular attention to the views of vulnerable and socially disadvantaged groups.

## Supporting information

**S1 Appendix. Interview guide.**
(PDF)

**S2 Appendix. Vignette.**
(PDF)

## Acknowledgments

The authors would like to thank all study participants for their time and valuable contributions to this project. The authors would also like to thank the Precise4Q consortium partners for their feedback on study design and support in recruitment.

## Author Contributions

**Conceptualization:** Julia Amann, Effy Vayena, Alessandro Blasimme.

**Data curation:** Julia Amann.

**Formal analysis:** Julia Amann, Effy Vayena, Kelly E. Ormond, Alessandro Blasimme.

**Funding acquisition:** Effy Vayena, Dietmar Frey, Alessandro Blasimme.

**Investigation:** Julia Amann, Effy Vayena, Kelly E. Ormond, Alessandro Blasimme.

**Methodology:** Julia Amann, Alessandro Blasimme.

**Project administration:** Julia Amann.

**Visualization:** Julia Amann.

**Writing – original draft:** Julia Amann.

**Writing – review & editing:** Julia Amann, Effy Vayena, Kelly E. Ormond, Dietmar Frey, Vince I. Madai, Alessandro Blasimme.

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
