## [Decision Letter · Decision Letter 0]

7 Jul 2022

PONE-D-22-09654Expectations and attitudes towards medical artificial intelligence: A qualitative study in the field of strokePLOS ONE

Dear Dr. Amann,

Thank you for submitting your manuscript to PLOS ONE. After careful consideration, we feel that it has merit but does not fully meet PLOS ONE’s publication criteria as it currently stands. Therefore, we invite you to submit a revised version of the manuscript that addresses the points raised during the review process.

Although the reviewers highlighted some positive aspects, they have major concerns that the authors need to carefully address before reconsidering this manuscript for publication. Please address in detail those comments that most concern the validity of the study proposed, including the consistency of what study participants might have grasped from the questions. 

We look forward to receiving your revised manuscript.

Kind regards,

Sara Rubinelli

Academic Editor

PLOS ONE

Journal Requirements:

Reviewers' comments:

Reviewer's Responses to Questions

**Comments to the Author**

1. Is the manuscript technically sound, and do the data support the conclusions?

Reviewer #1: Yes

Reviewer #2: Yes

2. Has the statistical analysis been performed appropriately and rigorously? 

Reviewer #1: N/A

Reviewer #2: N/A

3. Have the authors made all data underlying the findings in their manuscript fully available?

Reviewer #1: Yes

Reviewer #2: Yes

4. Is the manuscript presented in an intelligible fashion and written in standard English?

Reviewer #1: Yes

Reviewer #2: Yes

5. Review Comments to the Author

Reviewer #1: The work of Amann and colleagues describe the impact of artificial intelligence (AI) in medicine, especially in the field of stroke. The authors used a semi-structured interview approach to identify the relationship of stakeholders, such as stroke-survivors, their family and healthcare professionals, to the use of AI in a clinical context. The authors found that across study participants the use of AI is seen positively, however, they were hesitant if AI is used as a stand-alone, not supervised solution. Overall, especially the responsibility, legal and ethical aspect is discussed, and advocate that these issues need to be addressed in an appropriate fashion.

In general, as an AI practitioner, who is heavily involved in clinical practice and tries to bring sophisticated AI into the clinic, I really like the comprehensive view of the authors. However, I personally would like to rephrase the AI part or at least specify that the authors most likely mean deep learning/Neural networks with their black box character. I have a few comments on the technical slang.

Major comments

- Discussion that AI is commonly biased, racist and sexist.

Please discuss these aspects as I think this highly influences views, decisions, and may affect diagnoses and therapy. The point of view of the interviewed people that AI “may compensate for human bias” (l. 268) will definitely not happen without precautionary measures, and a subsentence like in l. 426 is not enough

- Some of the comments mention digitalization of health, such as repeatedly giving information, rather than focusing on AI. As the paper should highlight the AI, I would see that the authors do highlight AI-related comments.

Minor comments

- L. 83-86. The “if-then” statements do also apply for AI-based solutions. For example, decision trees are a very common example. Also, the expert medical knowledge is/or can be used to train an AI system. I think the authors mean that in deep learning features are extracted by the neural network, esp. In medical imaging approaches, where it is hard to explain/interpret how the system works.

- L. 89-90. In each ultrasound system to monitor any pregnancy AI is built in and used to date. Again, I think the authors mean latest deep neural networks.

- L. 102. Please emphasize the perceived risks. The use of medical AI is both, emotionally impacting the patient (“a machine is treating me”) and rationally impacting (“a machine given the circumstances, no matter of night, day or the 50th patient in a row gives me the same accuracy and quality”).

- L. 142. I was wondering how non-professionals deal with “ethical questions”. Even ethics experts have doubts about the use of AI and responsibility, I would like to see this discussed appropriately.

L. 175. Silly, but could you please rename “Findings” to “Results”? It is more convenient for academic papers.

- HCP. I assume that means Health Care Professional? Please introduce the abbreviation appropriately.

- L. 223-225. This sounds like the description of a digital twin (“what worked for person X with the same symptoms/stroke signs may also work for you”). I think the name dropping would be here appropriate.

- L. 525ff. I am not sure about these sentences. The problem of explainable AI is mostly because e.g. Deep Neural Networks are used for the analysis of unstructured data, but how they work is barely explainable. Whereas for structured data, models like support vector machines, logistic regression and decision trees are very easy to comprehend and interpretable. This is why particular focus is on the former instead of the latter.

Reviewer #2: AI in healthcare is a complicated issue that not only stroke survivors but also physicians may lack basic knowledge on the subject.

My main concern is that patients or health care providers could possibly be nudged towards certain answers in difficult questions when they lacked deep knowledge of the subject. In some responses, I think the way "AI" is used, probably refers to "automation" rather than AI; in other instance, a patient probably refers to electronic health files rather than AI. Line 436: HCP4 is probably the only specific reference to an AI system used by a participant; HCP4 refers to "CT monitoring" and goes on to express his concerns on automated arrhythmia detection... Isn't she/he referring to an ECG monitoring rather than a CT monitoring? Probably implantable Holter for paroxysmal AF detection.

The study has been promoted on websites in search of participants as research on AI (?), meaning participants would probably show interest in something they are already aware about. Had health care professionals used AI tools in their everyday clinical practice? Has AI been used in the care of stroke patients at any point? If so, were patients aware of the use of AI? If not, what exactly has been evaluated in this study? Their general knowledge on the subject?

Have the researchers used open-ended questions in the beginning of the interviews? what was the reaction of the researchers when answers revealed misunderstanding of what AI in stroke care means ? Did they provide information on what AI in stroke care means, before showing the clinical vignette?

Despite the fact that the manuscript is well-written, I find very little evidence that participants' knowledge and, therefore, views on AI would be any different from the general population of physicians and patients, since there seems to have little practical experience arising from true encounters with AI tools.

6. PLOS authors have the option to publish the peer review history of their article (what does this mean?). If published, this will include your full peer review and any attached files.

Reviewer #1: No

Reviewer #2: **Yes: **Dr Safouris Apostolos

---

## [Author Response · Author response to Decision Letter 0]

19 Aug 2022

Thank you very much for the opportunity to revise and resubmit our paper « Expectations and attitudes towards medical artificial intelligence: A qualitative study in the field of stroke». We greatly appreciate the reviewers' valuable comments which we believe helped to significantly improve the quality of our work. Please find below our point-by-point response to the reviewers’ comments.

We confirm that our manuscript meets PLOS ONE's style requirements. In line with PLOS ONE’s guidelines on data availability, we make translated excerpts of the transcripts relevant to the study available within the paper. 

Thank you very much for receiving our manuscript. We appreciate your time and look forward to hearing from you.

Sincerely,

Julia Amann 

Response to Journal Requirements:

Journal Requirements:

1. The manuscript meets PLOS ONE's style requirements.

2. Thank you for updating our Data Availability statement: “Translated excerpts of the transcripts relevant to the study are available within the paper.” Thank you for updating our data availability statement.

3. The ethics statement appears in the Methods section and has been removed from all other sections. 

Response to the reviewers’ comments:

Reviewer #1: 

The work of Amann and colleagues describe the impact of artificial intelligence (AI) in medicine, especially in the field of stroke. The authors used a semi-structured interview approach to identify the relationship of stakeholders, such as stroke-survivors, their family and healthcare professionals, to the use of AI in a clinical context. The authors found that across study participants the use of AI is seen positively, however, they were hesitant if AI is used as a stand-alone, not supervised solution. Overall, especially the responsibility, legal and ethical aspect is discussed, and advocate that these issues need to be addressed in an appropriate fashion.

In general, as an AI practitioner, who is heavily involved in clinical practice and tries to bring sophisticated AI into the clinic, I really like the comprehensive view of the authors. However, I personally would like to rephrase the AI part or at least specify that the authors most likely mean deep learning/Neural networks with their black box character. I have a few comments on the technical slang.

Response: Thank you very much for your thorough reading of our manuscript and your feedback. We particularly appreciate your comments regarding technical slang and have updated wording accordingly were appropriate. However, given that the term used during the interviews was “artificial intelligence” we are hesitant to change this in the manuscript, we did however add a note highlighting that we used AI as an umbrella term (See section on Data collection, L150-154).

Major comments

- Discussion that AI is commonly biased, racist and sexist.

Please discuss these aspects as I think this highly influences views, decisions, and may affect diagnoses and therapy. The point of view of the interviewed people that AI “may compensate for human bias” (l. 268) will definitely not happen without precautionary measures, and a subsentence like in l. 426 is not enough

Response: We fully agree with the reviewer that AI bias needs to be addressed. In the revised manuscript, we now address this issue in this discussion section (L. 539-541).

- Some of the comments mention digitalization of health, such as repeatedly giving information, rather than focusing on AI. As the paper should highlight the AI, I would see that the authors do highlight AI-related comments.

Response: Thank you for raising this point. We are aware that some of the participants’ comments relate to issues of data accessibility and digitalization of health rather than being solely focused on AI. However, in our view, these foundational issues form part of the larger discourse around the understanding of medical AI and should therefore not be dismissed. Moreover, since we were keen to explore participants’ views, we did not correct them when they talked about possible areas of applications or benefits that would, in fact, not depend on AI-based solutions. We now also clarify this in the revised version of the manuscript (L.145-149).

Minor comments

- L. 83-86. The “if-then” statements do also apply for AI-based solutions. For example, decision trees are a very common example. Also, the expert medical knowledge is/or can be used to train an AI system. I think the authors mean that in deep learning features are extracted by the neural network, esp. In medical imaging approaches, where it is hard to explain/interpret how the system works.

Response: Thank you for this comment, we agree that this was not well phrased. We have clarified in the respective sentence that we meant manually pre-defined sets of rules vs. the data-driven approach of machine learning based AI to learn from data. This characteristic is not restricted to neural nets but applies to all types of machine learning (L.83-85). 

- L. 89-90. In each ultrasound system to monitor any pregnancy AI is built in and used to date. Again, I think the authors mean latest deep neural networks.

Response: Thank you for this comment, we have included a clarifying statement in the introduction (L.89-90).

- L. 102. Please emphasize the perceived risks. The use of medical AI is both, emotionally impacting the patient (“a machine is treating me”) and rationally impacting (“a machine given the circumstances, no matter of night, day or the 50th patient in a row gives me the same accuracy and quality”).

Response: We feel that overemphasizing the perceived risks in the introduction would lead to imbalance. Moreover, both the emotional and rational impact of medical AI are addressed in the remainder of the paper.

- L. 142. I was wondering how non-professionals deal with “ethical questions”. Even ethics experts have doubts about the use of AI and responsibility, I would like to see this discussed appropriately.

Response: Thank you for raising this point. We did not expect participants to provide theoretically founded arguments but acknowledged that their responses would primarily stem from personal views and moral intuitions. We have added a clarifying statement (L.145-146).

L. 175. Silly, but could you please rename “Findings” to “Results”? It is more convenient for academic papers.

Response: We have renamed the section, it now reads “Results”. (L.185)

- HCP. I assume that means Health Care Professional? Please introduce the abbreviation appropriately.

Response: Thank you for raising this point, we have now introduced the abbreviation at the beginning of the results section and the Table 1. We have also indicated abbreviations indicating stroke survivors’ (Pat) and family members’ (FM) quotes. (L. 187-189; Table 1)

- L. 223-225. This sounds like the description of a digital twin (“what worked for person X with the same symptoms/stroke signs may also work for you”). I think the name dropping would be here appropriate.

Response: Thank you very much for pointing this out, we have now included a reference to digital twin technology, which we believe is indeed relevant to mentioned here. (L. 234)

- L. 525ff. I am not sure about these sentences. The problem of explainable AI is mostly because e.g. Deep Neural Networks are used for the analysis of unstructured data, but how they work is barely explainable. Whereas for structured data, models like support vector machines, logistic regression and decision trees are very easy to comprehend and interpretable. This is why particular focus is on the former instead of the latter.

Response: Thank you for raising this issue. We would like to point out, however, that the “vanilla” versions of many of the mention non-neural methods seldomly have sufficient performance. For example, with decision trees, ensemble methods like random forests (bagging) or boosted trees (boosting) are used, and these versions cannot be understood easily anymore. We have added a clarifying statement (L. 534-536).

Reviewer #2: 

AI in healthcare is a complicated issue that not only stroke survivors but also physicians may lack basic knowledge on the subject. My main concern is that patients or health care providers could possibly be nudged towards certain answers in difficult questions when they lacked deep knowledge of the subject. In some responses, I think the way "AI" is used, probably refers to "automation" rather than AI; in other instance, a patient probably refers to electronic health files rather than AI. Line 436: HCP4 is probably the only specific reference to an AI system used by a participant; HCP4 refers to "CT monitoring" and goes on to express his concerns on automated arrhythmia detection... Isn't she/he referring to an ECG monitoring rather than a CT monitoring? Probably implantable Holter for paroxysmal AF detection.

Response: Thank you for raising this important point, which we think very much reflects our findings, namely that there is in fact no common understanding of what medical AI is and what it can/cannot do among our study participants - and this is precisely one of the core issues we think needs to be addressed (see Discussion: Moving towards a more nuanced understanding of medical AI). To clarify this issue, we have also added a statement to the results section. (L. 204-205)

The study has been promoted on websites in search of participants as research on AI (?), meaning participants would probably show interest in something they are already aware about. 

Response: We fully agree with the reviewer and have addressed this point in our Strengths and limitations section: “We do have to consider the possibility of a selection bias, i.e., that our sample included more technology-savvy participants who may have had more favorable attitudes towards medical AI in stroke medicine.” (L. 622-624)

Had health care professionals used AI tools in their everyday clinical practice? 

Response: No, but some had read or heard about applications. We have added a clarifying statement in the results section (L. 189-191)

Has AI been used in the care of stroke patients at any point? 

Response: No, but some had read or heard about applications. We have added a clarifying statement in the results section (L. 189-191)

If so, were patients aware of the use of AI? If not, what exactly has been evaluated in this study? Their general knowledge on the subject?

Response: Thank you for raising this question. As noted in the paper, the primary aims of this study were to explore the views of stroke survivors, their family members, and healthcare professionals specialized in stroke regarding the use of stroke related medical AI. More specifically, we aimed to elicit their expectations and attitudes towards stroke related medical AI, focusing on the perceived benefits and risks when applied in the clinical setting. For this purpose, we presented them with a short vignette (Appendix 2), which described a concrete example of an AI-based CDSS as a prompt – all participants were presented with the same vignette to ensure they all responded to the same scenario. We can of course not exclude the possibility that their ideas may have shaped how they understood the example provided to them.

Have the researchers used open-ended questions in the beginning of the interviews? 

Response: Thank you for addressing this point. The interview guide consisted of three sections, each containing a set of open-ended question. Prompts and follow-up questions were used for deeper exploration/clarification, as it is customary in qualitative research. To clarify this issue, we have added the interview guide as an appendix. (Appendix 1)

What was the reaction of the researchers when answers revealed misunderstanding of what AI in stroke care means? 

Response: As we aimed to explore participants’ views and understanding of medical AI, we did not correct them as this would have a) biased our data and b) may have discouraged or intimidated participants to share their views in the fear of saying something wrong/incorrect. On the contrary, we tried to make participants as comfortable as possible to freely express their views, as it is customary in qualitative research. We have added a statement to clarify this issue. (L. 146-149)

Did they provide information on what AI in stroke care means, before showing the clinical vignette?

Response: Prior to showing the vignette, the interviewer gave a brief and simple explanation. To clarify this issue, we have added the interview guide as an appendix (Appendix 1)

Despite the fact that the manuscript is well-written, I find very little evidence that participants' knowledge and, therefore, views on AI would be any different from the general population of physicians and patients, since there seems to have little practical experience arising from true encounters with AI tools.

Response: We do not claim that the results of our study are generalizable, which is also not a goal of qualitative research, but if the reviewer considers them applicable more broadly, this actually speaks in favor of our study. As noted earlier, we were keen to explore participants’ views towards medical AI. For the purpose of this study and in line with earlier work, we conceptualized views to encompass attitudes, opinions, beliefs, feelings, understandings, experiences, and expectations. If we had been keen to focus on participants’ direct experiences with these technologies, we would have adopted an entirely different recruitment and data collection strategy. That being said, it is certainly a topic, we are keen to explore further and thank the reviewer for highlighting it.

---

## [Decision Letter · Decision Letter 1]

1 Dec 2022

Expectations and attitudes towards medical artificial intelligence: A qualitative study in the field of stroke

PONE-D-22-09654R1

Dear Dr. Amann,

We’re pleased to inform you that your manuscript has been judged scientifically suitable for publication and will be formally accepted for publication once it meets all outstanding technical requirements.

Kind regards,

Federica Canzan

Academic Editor

PLOS ONE

Reviewer's Responses to Questions

**Comments to the Author**

1. If the authors have adequately addressed your comments raised in a previous round of review and you feel that this manuscript is now acceptable for publication, you may indicate that here to bypass the “Comments to the Author” section, enter your conflict of interest statement in the “Confidential to Editor” section, and submit your "Accept" recommendation.

Reviewer #1: All comments have been addressed

Reviewer #2: All comments have been addressed

2. Is the manuscript technically sound, and do the data support the conclusions?

Reviewer #1: Yes

Reviewer #2: Yes

3. Has the statistical analysis been performed appropriately and rigorously? 

Reviewer #1: N/A

Reviewer #2: N/A

4. Have the authors made all data underlying the findings in their manuscript fully available?

Reviewer #1: No

Reviewer #2: Yes

5. Is the manuscript presented in an intelligible fashion and written in standard English?

Reviewer #1: Yes

Reviewer #2: Yes

6. Review Comments to the Author

Reviewer #1: Please provide a respective link to the minimal dataset to reproduce your findings or a dedicated zenodo/Figshare repository to ensure data availability, which is not by request to the authors only.

Reviewer #2: All issues have been adequately addressed, I have no further comments, this is a well-written manuscript and adresses the research question with a sound methodology.

7. PLOS authors have the option to publish the peer review history of their article (what does this mean?). If published, this will include your full peer review and any attached files.

Reviewer #1: No

Reviewer #2: **Yes: **Dr Safouris Apostolos

---

## [Editor Report · Acceptance letter]

20 Dec 2022

PONE-D-22-09654R1 

Expectations and attitudes towards medical artificial intelligence: A qualitative study in the field of stroke 

Dear Dr. Amann:

I'm pleased to inform you that your manuscript has been deemed suitable for publication in PLOS ONE. Congratulations! Your manuscript is now with our production department. 

Kind regards, 

on behalf of

Professor Federica Canzan 

Academic Editor

PLOS ONE